# Information Dynamics of Electric Field Intensity before and during the COVID-19 Pandemic

**DOI:** 10.3390/e24050726

**Published:** 2022-05-20

**Authors:** Gorana Mijatovic, Dragan Kljajic, Karolina Kasas-Lazetic, Miodrag Milutinov, Salvatore Stivala, Alessandro Busacca, Alfonso Carmelo Cino, Sebastiano Stramaglia, Luca Faes

**Affiliations:** 1Faculty of Technical Sciences, University of Novi Sad, 21102 Novi Sad, Serbia; gorana86@uns.ac.rs (G.M.); dkljajic@uns.ac.rs (D.K.); kkasas@uns.ac.rs (K.K.-L.); miodragm@uns.ac.rs (M.M.); 2Department of Engineering, University of Palermo, 90128 Palermo, Italy; salvatore.stivala@unipa.it (S.S.); alessandro.busacca@unipa.it (A.B.); alfonsocarmelo.cino@unipa.it (A.C.C.); 3Department of Physics, University of Bari, 70121 Bari, Italy; sebastiano.stramaglia@ba.infn.it

**Keywords:** dynamical systems, electric field intensity, nonlinear dynamics, predictability, complexity, human mobility

## Abstract

This work investigates the temporal statistical structure of time series of electric field (EF) intensity recorded with the aim of exploring the dynamical patterns associated with periods with different human activity in urban areas. The analyzed time series were obtained from a sensor of the EMF RATEL monitoring system installed in the campus area of the University of Novi Sad, Serbia. The sensor performs wideband cumulative EF intensity monitoring of all active commercial EF sources, thus including those linked to human utilization of wireless communication systems. Monitoring was performed continuously during the years 2019 and 2020, allowing us to investigate the effects on the patterns of EF intensity of varying conditions of human mobility, including regular teaching and exam activity within the campus, as well as limitations to mobility related to the COVID-19 pandemic. Time series analysis was performed using both simple statistics (mean and variance) and combining the information-theoretic measure of information storage (IS) with the method of surrogate data to quantify the regularity of EF dynamic patterns and detect the presence of nonlinear dynamics. Moreover, to assess the possible coexistence of dynamic behaviors across multiple temporal scales, IS analysis was performed over consecutive observation windows lasting one day, week, month, and year, respectively coarse grained at time scales of 6 min, 30 min, 2 h, and 1 day. Our results document that the EF intensity patterns of variability are modulated by the movement of people at daily, weekly, and monthly scales, and are blunted during periods of restricted mobility related to the COVID-19 pandemic. Mobility restrictions also affected significantly the regularity of the EF intensity time series, resulting in lower values of IS observed simultaneously with a loss of nonlinear dynamics. Thus, our analysis can be useful to investigate changes in the global patterns of human mobility both during pandemics or other types of events, and from this perspective may serve to implement strategies for safety assessment and for optimizing the design of networks of EF sensors.

## 1. Introduction

The presence of electromagnetic fields (EMFs) arising from both natural and artificial sources represents a ubiquitous phenomenon in the human environment [1]. The natural origins of EMFs, such as the Sun, the Earth’s natural electric and magnetic field, cosmic radiations, and atmospheric discharges, were the only sources during the longest period of evolutionary development. The 20th century brought a sudden expansion of artificial (human-made) field sources, which are nowadays mostly due to the wireless transmission technology (WTT) [2]. Playing a key role in today’s communication, the WTT represents a wireless modality for data transfer by means of electromagnetic waves, and has become of huge practical importance in our daily activities. As such, this modality is central to emerging technologies over the next years, including 5G/6G cellular systems [3,4], new medical devices [5], robots and drones [6], self-driving vehicles [7], etc. By increasing the number of EMF sources, the intensity of EMF also increases. This increase can be induced by higher power transmitters (radio and TV transmitters, mobile phone base stations), as well as by the low-power devices that we use every day, such as mobile phones, microwave ovens, Wi-Fi routers, etc. An unavoidable effect of these phenomena is the chronic exposure of the population to EMF values, to an extent such that the exposure on wireless communication networks is often cited as a major cause of public concern about possible adverse effects of EMFs on health [8]. Accordingly, the scientific studies of EMF intensity treat this aspect as one of the most crucial, including recommendations by national and international health/scientific authorities, such as the International Commission on Non-Ionizing Radiation Protection, ICNIRP [9], for establishing safety limits for exposure to EMFs.

A relevant aspect in the study of EMFs, besides measuring the average intensity over a certain period, refers to the temporal variability of the electric field (EF) monitored during that period. A relevant question, raised in this work, is how the presence/absence or temporary gathering of people who use wireless communication services affects the variability over time of the EF intensity. To address this question, we perform a thorough analysis of the temporal statistical structure of time series reflecting cumulative intensity of wideband EFs. The time series are acquired by a sensor placed in the campus area of the University of Novi Sad, Serbia, during two consecutive years, 2019 and 2020. Analyzing data from this recording site allows us to investigate the underlying intrinsic dynamics of EF intensity depending on human mobility and gathering. The analysis of human mobility as a dynamic phenomenon has been extensively performed in the last two decades by interdisciplinary studies aimed to understand the intrinsic properties of human movements and the mechanisms behind the detected patterns [10,11], and to determine the effects of these patterns on the structure and dynamics of social networks [12,13], on economic phenomena [14], and recently even on the transmission of the COVID-19 disease [15,16]. While human mobility is typically analyzed by the direct use of cellular phone data through the global navigation satellite system (GNSS), here we deal with this concept (more precisely, with people absence/presence or gathering in the location of interest) implicitly, by observing its overall effects on the time series of EF intensity. Specifically, the availability of longitudinal data collected continuously over two years in the observed location offers the possibility to explore the effects of human presence and its extent on the dynamic changes of the EF intensity. These changes are related to contributions arising from EF sources linked to human utilization of wireless communication systems and the corresponding services, and thus have the potential to provide additional and complementary information to the one which can be gathered by GNSS data.

The dynamic patterns of EF intensity measured by the considered sensor capture different scenarios of human mobility, including the daily movements of university staff and students within the campus, the decreased mobility during weekends, the less regular gathering of people in the periods of university exams, the considerably reduced presence of university population during holidays and summer break, and—in the analyzed years—the blocked or restricted mobility of the university population due to the state of emergency related to the COVID-19 pandemic. To properly investigate this complex process featuring dynamic patterns deployed across multiple temporal scales, we use both simple statistics (mean, variance) and information-theoretic measures of dynamical complexity implemented through a multiscale approach. Specifically, we characterize the EF intensity time series using the information storage (IS), a measure quantifying the amount of uncertainty about the present state of a dynamic system that can be resolved by the knowledge of its past states [17]. Recognized as one of the three key components of information processing (i.e., information storage, transfer, and modification) [18], IS reflects the regularity of the dynamic patterns in a time series, and has been applied with broad relevance in diverse real-world dynamical systems ranging from human–brain networks [19] and robot motion [20] to cardiorespiratory [21] and cardiovascular [22] dynamics. The popularity of dynamical information measures like IS stems primarily from their generality and applicability to short-length and noisy processes, which allow for characterizing the dynamical nature of the systems generating the observed time series in an efficient and robust way. Besides studying the temporal variability of the EF intensity and its regularity, in this work, we combine the estimation of IS [23] with the method of surrogate data [24], to investigate the presence of nonlinear dynamics in such time series. In addition, to account for the possible coexistence of oscillatory behaviors at different temporal scales, we follow a multi-scale approach [25], whereby IS is computed after “rescaling” the observed process to focus on a specific range of temporal scales; rescaling is performed by filtering out the fast temporal features from the observed EF time series before the computation of IS. The resulting time series, mapping the average EF intensity collected at time scales of 6 min, 30 min, 2 h, and one day, and analyzed respectively over epochs lasting one day, one week, one month, and one year, are investigated to capture the multiscale complexity and nonlinearity of the dynamic changes of EF exposure before and during the COVID-19 pandemic.

## 2. Materials and Methods

This section presents the data used in the work and describes the statistical measures used to analyze them. Our analysis is focused on time series of EF intensity whose temporal statistical properties are assessed by the information-theoretic measure of *information storage* (IS). The IS measure is computed to quantify the regularity of the analyzed EF intensity time series, i.e., the presence of repetitive (predictable) patterns in the series. Regularity measures like IS are inversely related to the *complexity* of the time series intended as its degree of unpredictability. In addition, the method of surrogate data associated with the quantification of regularity is employed to investigate the presence of *nonlinearity* in the analyzed time series, intended as the existence of nonlinear dynamics: a time series is regarded as nonlinear if its regularity quantified by the IS measure is significantly larger than the regularity of surrogate series in which the linear temporal correlations are preserved, but nonlinear correlations are destroyed. Note that the concepts of complexity and nonlinearity do not necessarily align: a time series can be very complex (thus barely predictable) even if it is governed by linear dynamics, or very predictable (thus with small complexity) even if it is determined by nonlinear relations. Our analyses are focused on establishing separately, respectively using the IS and the method of surrogate data, the level of complexity of the EF time series, and the existence of nonlinear dynamics during different observation windows.

### 2.1. EMF RATEL Monitoring System and EF Intensity Time Series

The data analyzed in this work originate from the wireless sensors network launched in 2017 by the Serbian Regulatory Agency for Electronic Communications and Postal Services (RATEL). The network, called EMF RATEL, performs day-to-day monitoring of EF intensity and assessment of the overall population exposure to the field values [26]. The system is designed to operate in heterogeneous network environments, and incorporates various classes of EMF monitoring sensors which perform frequency-selective or broadband cumulative measurements of the EF level, collecting signals from all the active EF sources which surround the sensors placed in a given location [27]. The network currently contains 88 wireless sensors strategically placed in zones of high sensitivity (i.e., where people are expected to stay for longer periods, such as universities, schools, and hospitals) across the whole territory of the Republic of Serbia [26]. The field probes of the used sensors are isotropic, which means that the monitoring is performed by considering all field components. The average EF intensity measured by the sensors over consecutive periods of six minutes is stored in the form of time series data and transferred at a daily basis to the centralized database of the EMF RATEL system [27].

The time series analyzed in this study were acquired by a sensor providing broadband cumulative EF intensity measurements in the frequency range 100 kHz–7 GHz [27], located in the campus area of the University of Novi Sad, Serbia. This location is characterized as a densely populated area in regular circumstances, i.e., during semesters and exams periods, and with a considerably reduced circulation of university population during the summer break in August. The data used in this work were acquired during 2019 and 2020, so as to cover the activity collected in the campus before and during the pandemic of COVID-19. These data provided by RATEL, as well as the data from all other locations where sensors are installed, are available and can be freely downloaded from the Serbian Open Data Portal [28]; the data set used here can be downloaded from the link [29].

### 2.2. Time Series Pre-Processing

The EF intensity time series recorded by the sensor described above were treated as realizations of a discrete-time process mapping the dynamics activity of the EF monitoring station. In this work, motivated by the knowledge that dynamical systems typically exhibit a multi-scaled complexity as they operate across multiple spatial and temporal scales, before performing statistical analyses, we pre-processed the recorded time series according to the *coarse graining* procedure described in the following [25].

First, starting from the time series collected with a sampling period of six minutes (sampling frequency $f_{s}=0.167$ min${}^{-1}$), we considered portions of the time series covering the EF intensity over four different observation windows, i.e., at each day (N=240 samples), week (N=1680 samples), month (N=7200 samples), and year (*N* = 86,400 samples). The portion of the time series corresponding to each observation window was coarse-grained by averaging its values over consecutive non-overlapping sequences of *q* samples. Different values of *q* are used depending on the duration of the observation window, generating different time scales of observation: q=1 for windows covering 1 day, so as to set a time scale $\tau_{day}$ = 6 min which retains the originally measured EF activity; q=5 for windows covering 1 week, so as to set a time scale $\tau_{week}$ = 30 min; q=20 for windows covering 1 month, so as to set a time scale $\tau_{month}$ = 2 h; and q=240 for windows covering the whole year, so as to set a time scale $\tau_{year}$ = 1 day. Using these time scales resulted in observation windows of similar length, i.e., $N_{\left(q=1\right)}\approx 240$ samples, $N_{\left(q=5\right)}\approx 336$ samples, $N_{\left(q=20\right)}\approx 360$ samples, and $N_{\left(q=240\right)}\approx 365$ samples. Since averaging is a form of low-pass filtering, the effect of the coarse-graining procedure is to cut off from the dynamics the fast temporal scales, i.e., those associated with oscillations having frequency $>f_{c}=\frac{f_{s}}{2q}$[25,30]; in our case, setting q=5, q=20 and q=240 means that the one-week windows focus on oscillations with frequency lower than $f_{c}$ = 0.0167 min${}^{-1}$, the one-month windows focus on oscillations with frequency lower than $f_{c}$ = 0.0042 min${}^{-1}$, and the one-year windows focus on oscillations with frequency lower than $f_{c}$ = 0.00034 min${}^{-1}$.

The time series re-scaled as described above were first analyzed using simple statistics such as the mean and the variance, in order to retrieve information about the average value and the overall variability within each observation window. Afterwards, the measure of information storage and the method of surrogate data, described in the following section, were applied. Before information-theoretic analysis, the re-scaled time series was detrended to remove slow variations that can have a detrimental effect on the computation of entropy measures [23]. This step was performed by filtering the coarse-grained time series by a linear high-pass filter (cutoff frequency at 3 dB: 0.0264 cycle${}^{-1}$ [31]).

### 2.3. Information-Theoretic Analysis

The multiscale regularity of the time series of EF intensity was quantified in the information-theoretic domain using the measure of information storage. This measure investigates for the discrete-time stochastic process *X* describing the EF dynamics rescaled according to the procedure described in Section 2.2, the statistical dependencies between the scalar random variable $X_{n}$ sampling the process at the present time *n* and the vector random variable $X_{n}^{m}=\left[X_{n-1},\ldots ,X_{n-m}\right]$ sampling its past collected over *m* time lags. The IS is defined as the mutual information (MI) between the present state and the whole past history of the process *X* [17,23]:(1)$$IS\left(X\right):=\lim_{m\to \infty }I\left(X_{n};X_{n}^{m}\right),$$
where the MI between the present and the past *m* states is defined as [32]:(2)$$I\left(X_{n};X_{n}^{m}\right)=E\left[\log\frac{p(x_{n},x_{n}^{m})}{p\left(x_{n}^{m}\right)p\left(x_{n}\right)}\right],$$
where $p(x_{n})$ and $p(x_{n}^{m})$ are the probabilities that the process *X* takes the value $x_{n}$ at the current time *n* and the values $x_{n-1},\ldots ,x_{n-m}$ over the past *m* time lags, respectively, and E[·] denotes the expectation operator. The IS measure quantifies the amount of information about the present state of the process that can be explained by its own past states. As such, it reflects the *regularity* of the process dynamics intended as their predictability: if the process is fully random, the past does not bring any knowledge about the present and IS = 0; on the contrary, if the process is fully predictable, the present can be entirely predicted from the past and IS takes its maximum value equal to entropy of the process states, $H\left(X\right):=H\left(X_{n}\right)=-E\left[\log p\left(x_{n}\right)\right]$ [23].

From the point of view of the dynamic update of the system states, the information storage is complementary to a well-known measure of complexity, quantified in terms of the conditional entropy (CE) of the present state of the process given its past states, $CE\left(X\right):=\lim_{m\to \infty }H\left(X_{n}|X_{n}^{m}\right)$, where $H\left(X_{n}|X_{n}^{m}\right)=-E\left[\log p\left(x_{n}|x_{n}^{m}\right)\right]$, with $p\left(x_{n}|x_{n}^{m}\right)=p\left(x_{n},x_{n}^{m}\right)/p\left(x_{n}^{m}\right)$ being the conditional probability of $x_{n}$ given $x_{n}^{m}$; in fact, it can be easily seen that the CE and the IS of a process are related to each other by the simple relation H(X)=IS(X)+CE(X) [23]. Thus, the multiscale analysis of IS performed in this study evaluates the regularity of the process dynamics across multiple temporal scales [33], and provides complementary information to the multiscale analysis of CE performed by the so-called multiscale entropy method [25].

### 2.4. Estimation of the Information Storage

This section describes the estimation approach used to compute the IS measure defined in Section 2.3. Dynamic information measures like the IS are typically computed approximating the past history of the analyzed process with a finite number of lagged components, i.e., using Equation (Equation 2) to estimate Equation (Equation 1). In our work, we used *m* = 2 samples to cover the past history of the process, as it is usually done for short datasets [34,35]. Then, an estimator of the MI in Equation (Equation 2) is implemented to compute the IS for each rescaled time series. However, different approaches are available for the estimation of information dynamic measures [23] and multiscale computation of IS has been proposed in the context of linear parametric estimation [33]; in this work, we adopt the nonlinear model-free approach proposed in [21,23], which brings the advantage of allowing the detection of potentially any form of predictable dynamics within the analyzed process. This approach makes use of the *k*-nearest neighbor (knn) estimator of information measures, which is an asymptotically unbiased and consistent estimator for computing the entropy of a multidimensional random variable based on the statistics of the distances between realizations of the variable [36].

Specifically, the knn estimator computes the differential entropy of a generic *d*-dimensional variable *W* as:(3)$$\hat{H}\left(W\right)=-\psi \left(k\right)+\log\left(N^{\prime }-1\right)+\log c_{d,L}+\frac{d}{N^{\prime }}\sum_{n=1}^{N^{\prime }}\log\epsilon_{n,k},$$
where ψ(·) is the digamma function, $c_{d,L}$ is the volume of the *d*-dimensional unit ball under a given norm *L* ($c_{d,L}=1$ for the maximum norm used in this work which takes the maximum distance of the scalar components), $\epsilon_{n,k}$ is twice the distance between the $n^{th}$ observation of *W* and its $k^{th}$ nearest neighbor, and $N^{\prime }$ is the number of observations available [36]. Note that, once the free parameter *k* is chosen, the estimation of $\hat{H}\left(W\right)$ is performed through a neighbor search around each data point which returns the distances $\epsilon_{n,k}$ to be plugged in Equation (Equation 3). This entropy estimator is adopted to compute the three entropy terms that compose the IS, which is then estimated as:(4)$$\hat{I}\left(X_{n};X_{n}^{m}\right)=\hat{H}\left(X_{n}\right)+\hat{H}\left(X_{n}^{m}\right)-\hat{H}\left(X_{n},X_{n}^{m}\right).$$However, instead of naively applying Equation (Equation 3) three times performing neighbor searches in the spaces spanned by the observations of $X_{n}$, $X_{n}^{m}$ and $\left(X_{n},X_{n}^{m}\right)$, in this work, we adopt the procedure first proposed in Ref. [37], which allows for compensating the bias arising from the different dimension of the three entropy terms. This procedure is based on performing a neighbor search in the space spanned by the observations of the highest-dimensional variable to find the distances between any observation and its $k^{th}$ neighbor, and then using these distances in the projected lower-dimensional spaces as the range inside which the neighbors are counted. More specifically, the knn estimate of the entropy of the highest-dimensional variable is computed through neighbor search: (5)$$\hat{H}\left(X_{n},X_{n}^{m}\right)=-\psi \left(k\right)+\log\left(N-m-1\right)+\frac{m+1}{N-m}\sum_{n=1}^{N-m}\log\epsilon_{n,k},$$where $\epsilon_{n,k}$ is twice the distance from $n^{th}$ observation of $\left(X_{n},X_{n}^{m}\right)$ to its $k^{th}$ neighbor.

Then, knowing the distances $\epsilon_{n,k}$, the entropies of the lower-dimensional variables are estimated through two separate range searches:
(6)$$\hat{H}\left(X_{n}^{m}\right)=-\frac{1}{N-m}\sum_{n=1}^{N-m}\psi \left(N_{X_{n}^{m}}\right)+\log\left(N-m-1\right)+\frac{m}{N-m}\sum_{n=1}^{N-m}\log\epsilon_{n,k},$$
(7)$$\hat{H}\left(X_{n}\right)=-\frac{1}{N-m}\sum_{n=1}^{N-m}\psi \left(N_{X_{n}}\right)+\log\left(N-m-1\right)+\frac{1}{N-m}\sum_{n=1}^{N-m}\log\epsilon_{n,k},$$where $N_{X_{n}^{m}}$ and $N_{X_{n}}$ are the number of points whose distance from $n^{th}$ observation of $X_{n}^{m}$ and $X_{n}$, respectively, is smaller than $\epsilon_{n,k}/2$.

Finally, substituting Equations (Equation 5)–(Equation 7) into Equation (Equation 4) yields the knn estimate of the IS:
(8)$$\hat{I}\left(X_{n};X_{n}^{m}\right)=\psi \left(k\right)+\log\left(N-m-1\right)-\frac{1}{N-m}\sum_{n=1}^{N-m}\left(\psi \left(N_{X_{n}}\right)+\psi \left(N_{X_{n}^{m}}\right)\right).$$

In this work, the estimation of IS was performed by setting the estimation free parameter at the value *k* = 10 neighbors, which has been shown to achieve a good trade-off between bias and variance of the estimates [23].

### 2.5. Surrogate Data Analysis

In this work, the existence of a statistically significant degree of information storage and the presence of nonlinear dynamics in the EF intensity time series were tested using the method of surrogate data [24,38]. This method tests a given property of the data under analysis first specifying a null hypothesis about some properties of the analyzed data (in our case that the underlying process is random, or that it possesses linear correlations only); the null hypothesis can be rejected or accepted by the statistical test. Second, a set of surrogate time series which satisfies the null hypothesis but not the property under evaluation is built. Third, a discriminating statistic (in our case, the information storage) is computed both on the original time series and over the distribution of the surrogate series. Finally, a statistical test which compares the discriminating statistic computed on the original series with its distribution on the surrogates is performed, which allows for detecting the presence of the investigated property if the null hypothesis is rejected, or the absence of the property if the null hypothesis is not rejected [24]; in this work, we did always generate 100 surrogate series for each original analyzed EF time series, and selected a statistical significance α=0.01 for the hypothesis test.

To test the presence of a significant amount of information stored in the analyzed EF time series, we set the null hypothesis that the series is a realization of a serially uncorrelated process with the same probability distribution of the original process. Then, the surrogate time series were generated by shuffling randomly the order of the samples in the original time series, so as to preserve the amplitude distribution and the entropy, while destroying any temporal relation between the samples (and thus also the information storage). The selected test was a one-tailed non-parametric test based on percentiles, whereby the original IS value computed individually for each time series was deemed as statistically significant if it exceeded the $100{\left(1-\alpha \right)}^{th}$ percentile of its distribution evaluated from the surrogate series.

To test the presence of nonlinearity in the dynamics of the analyzed EF time series, we set the null hypothesis that the series is a realization of a Gaussian stochastic process fully described by linear temporal correlations. In this case, the surrogate time series were generated imposing both the autocorrelation structure and the marginal distribution of the original time series. The autocorrelations in the original time series were maintained preserving the modulus of the power spectral density of the series while destroying the phase, while the amplitude distribution was preserved exploiting a rank-ordering procedure. This was achieved through the iteratively refined amplitude adjusted Fourier Transform (IAAFT) method [39], a procedure that alternatively constrains the surrogate series to have the same power spectrum (by imposing the squared Fourier amplitudes while randomizing the Fourier phases) and to have the same amplitude distribution (by rank ordering) of the original series. The statistical test was again a non-parametric test based on percentiles, whereby the IS estimated on the original EF intensity time series was compared with the IS distribution assessed over the surrogate series; the null hypothesis was rejected, thus detecting the presence of nonlinear dynamics, when the original IS was larger than the $100\cdot {\left(1-\alpha \right)}^{th}$ percentile of the distribution of IS computed on the surrogates, while it was accepted otherwise, thus deeming the observed dynamics as compatible with a linear Gaussian process [40].

## 3. Results and Discussion

Figure 1 shows a representative time series of the EF intensity observed during a representative day (a), week (b), and month (c) of 2019 and 2020, as well as during the whole year (d). We selected the 79th day, the 15th week, and the fourth month of 2019 and 2020, to compare the EF intensities before and during the COVID-19 pandemic; the series shown in Figure 1a–c is rescaled according to the coarse-graining procedure described in Section 2.2, to demonstrate the oscillations dominant in daily, weekly, and monthly observation windows.

Differences between the behavior of the EF intensity during the same period of the two years can be noticed immediately through simple visual inspection. The time series measured in 2019 (orange curves) clearly demonstrate a modulation of the EF intensity which follows the day-to-night cycle (Figure 1a,b, left) and a reduction in amplitude during the weekends (Figure 1c, left); we ascribe such modulations to the higher or lower amount of people present in the University area. When the same periods of 2020 are considered (green curves), the day-to-night variations are damped (Figure 1a,b, right) and do not differ anymore between weekdays and weekends (Figure 1c, right). We note that the difference between the EF intensities of the same periods of the two years appears evident by looking at the amplitude of the oscillations, but not at the mean values which are roughly the same. Moreover, these differences are not straightforward when the whole-year time series are observed (orange curve vs. green curve in Figure 1d).

To make the observations above quantitative and confirm them in a complete analysis, we report in Figure 2 and Figure 3 the mean $\mu_{EF}$ and the variance $\sigma_{EF}^{2}$ of the EF intensity observed at the four time scales and computed for each observation window during the two years. While the mean values of the EF intensity measured in 2019 and 2020 are quite similar (Figure 2d: $\mu_{EF}$ = 1.16 V/m in 2019 and 1.26 V/m in 2020), the dynamic behavior of the mean EF intensity emerges, analyzing the rescaled time series (Figure 2a–c), which display values oscillating between ∼0.2 V/m and ∼1.9 V/m. The trends of the mean EF intensity show oscillatory activity with a period of one week at the finer time scale (Figure 2a), as well as slower variations demonstrated when the time scale becomes more coarse (Figure 2b,c). Given that the oscillations with one-week period observed in the two years (green and orange curves in Figure 2a) exhibit minimum EF intensity during weekends, they can be related to the amount of people present on the university campus. However, when we compare 2019 and 2020, the averaged EF intensity is different between periods of the two years, which are expected to be similar in terms of human presence (e.g., the first three months where $\mu_{EF}$ is higher in 2019, the last three months where $\mu_{EF}$ is higher in 2020, and the periods of exams in person and summer pause where $\mu_{EF}$ is again higher in 2020). This effect is clearly evident observing also the slower trends in Figure 2b,c, thus reflecting the contributions from all detected EF sources in the proximity of an installed sensor. Nevertheless, an effect of the pandemic on the mean EF values occurs in correspondence with the State of Emergency (SoE) declared by the Government of Serbia on 15 March 2020 [41]: starting from this date, wherein strict movement restrictions were implemented, a sudden drop of $\mu_{EF}$ is visible at the daily and weekly observation windows (Figure 2a,b), and the reduced mean field value of ∼1.3 V/m is maintained during the spring and summer 2020, when a clear reduction in the amplitude of oscillations with one-week period is also observed (light orange, green and grey areas in Figure 2a).

The most distinctive feature that differentiates the mean EF intensity measured during the pandemic and in the preceding year without mobility restrictions is the dampening of the oscillations with a period of one week (Figure 2a, color-shaded areas). This behavior, suggesting that reduced mobility decreases the variability of the EF intensity more than its mean, becomes straightforward when the variance of the EF time series is analyzed (Figure 3). In fact, the analysis of EF variability documents very clearly that the variance $\sigma_{EF}^{2}$ drops dramatically (from 0.2–0.3 V${}^{2}$/m${}^{2}$ to almost zero) when the SoE is declared, and remains at very low values for the whole period of mobility restrictions when University teaching shifted to the online modality (Figure 3a,b). The two periods of University exams are characterized by very comparable values of $\sigma_{EF}^{2}$ in 2019 and 2020, which is in line with the fact that, in Serbia, the exams were held in person even during the pandemic. During the summer pause in August, the variance of the EF intensity was comparably very small in 2019 and 2020, while differences arise again during the fall when teaching was ordinarily in person in 2019 and again in online modality in 2020. These patterns of variability of the EF intensity are well-characterized at all time scales, as they are visible over observation windows of one day (Figure 3a), one week (Figure 3b), and one month (Figure 3c). The overall result is that the variability estimated on a yearly basis is lower in 2020 compared to 2019 (Figure 3d).

The results presented above indicate that the restrictions to mobility imposed by the COVID-19 pandemic (SoE, 2020) or the substantial absence of University population during the summer break (during both 2019 and 2020) affected primarily the variability of the electric fields measured by the sensor of the EMF RATEL system, and only to a lower extent the average field intensity. Here, it is worth noting that the monitoring sensor is installed on the roof of the Faculty of Technical Sciences (FTS), placed in the campus area of University of Novi Sad (the images following the sensor position can be found in the link [42]). Besides the FTS, the campus area encompasses six faculties more, while the FTS presents the largest one by the number of students. However, since the substantial reduction in the presence of people during the pandemic and summer break in such highly-populated area during the regular university life (teaching and exam periods) affected dominantly the variability of EF intensity series, to investigate the changes that originate this variability and its modification in different conditions, we move to study the storage of predictable information within the EF intensity time series observed at different time scales.

Figure 4 depicts the results the analysis of information storage performed for the representative time series reported in Figure 1, pre-processed as explained in Section 2.2. All the estimates of IS resulted n being statistically significant according to the test based on randomly shuffled surrogate data, meaning that the null hypothesis of a serially uncorrelated process was rejected for all the series in Figure 1 (data not shown). The amount of information stored in the EF dynamics is always higher for the exemplary time series measured during 2019 than for the series measured in the same period of 2020, with values $IS_{2019}$ = 0.42 nats and $IS_{2020}$ = 0.2 nats for the 79th day (Figure 4a), $IS_{2019}$ = 1.45 nats and $IS_{2020}$ = 0.62 nats for the 15th week (Figure 4b), $IS_{2019}$ = 1.33 nats and $IS_{2020}$ = 0.89 nats for the 4th month (Figure 4c), and $IS_{2019}$ = 0.65 nats and $IS_{2020}$ = 0.44 nats for the whole year (Figure 4d). The lower values of IS reflect higher complexity of the EF time series, documenting the presence of less predictable, more erratic EF dynamical patterns when human mobility was strictly restricted, i.e., during 2020 due to COVID-19 pandemic, in contrast to one year before when the higher IS values indicate the presence of more regular patterns of EF intensity. Nonetheless, the analysis based on IAAFT surrogate data reveals for all time scales the presence of nonlinear EF dynamics during 2019 (the original IS value exceeds the IS surrogate threshold in Figure 4a–d), while, on the contrary, the daily, weekly, and monthly dynamics observed during the same periods in 2020 can be regarded as linear (the original IS value is below the surrogate threshold in Figure 4a–c). This result distinguishes the nature of oscillations during the periods of normal and restricted mobility, also showing how complexity and nonlinearity are different concepts: in this case, the less complex dynamics (showing higher IS) are those with a stronger contribution of nonlinearities (IS is higher than the nonlinearity threshold).

The complete multiscale analysis was performed to investigate the trends of the information stored in the time series of EF intensity analyzed separately during 2019 and 2020 over daily, weekly, and monthly observation windows as well as over the whole year. According to the surrogate data analysis based on random shuffling, the null hypothesis of white noise process was rejected for all cases, thus assessing statistically the significance of the IS values in all the analyzed time series (data not shown). The detection of statistically significant multiscale information storage is in line with the existence of predictable dynamics in the series measured at different scales, indicating that the EF intensity is a complex but structured process characterized by regular oscillations across multiple scales of observation, which reflect an alternating presence of EF sources presumably related to human utilization of wireless communication services.

The values of information storage measured over observation windows of different length, each characterized by their peculiar time scale, are reported in Figure 5. The detailed analysis comparing the two considered years of EF recordings sheds further light on the nature of these predictable patterns. In fact, as a consequence of the onset of the COVID-19 pandemic in March 2020 and of the establishment of strict mobility restrictions, the EF intensity time series displays a significant reduction in the values of IS. The reduced regularity compared with the same period of the preceding year is visible in the EF dynamics observed on a daily basis (Figure 5a, light orange areas) as well as when investigating longer time-scales which cover observation windows of one week (Figure 5b, light orange areas) or of one month (Figure 5c, months 3, 4, 5). The presence of more erratic multiscale patterns of EF intensity is associated with the significantly reduced access of people to the university campus in this period when teaching lessons were delivered in the online modality.

At the highest level of coarse graining, the difference between dynamic behavior of the EF intensity measured during the two years is documented by the lower regularity measured during 2020 than in 2019 ( Figure 5d: $IS_{2019}$ = 0.65 nats and $IS_{2020}$ = 0.44 nats). On the other hand, the information stored in the EF intensity time series takes comparable values in the two years during the periods of university exams and summer pause (light green and grey areas in Figure 5a,b and months 6–9 in Figure 5c). In this time frame, the IS values are relatively high during the exam periods and decrease to lower values during the summer, suggesting that the predictability of the field intensity dynamics arises from regular movement of people within the area covered by the EF sensor. The similarity between the two years can be explained considering that the campus was populated in a similar way as exams were always delivered in person during the pandemic. Then, the prosecution of the online teaching modality during the fall/winter semester of 2020 determined lower and less regular EF dynamics compared to the fall/winter of 2019; again, the phenomenon is visible at all time scales (light yellow area in Figure 5a,b and months 10–12 in Figure 5c).

As regards the nature of the oscillatory patterns underlying the observed dynamics, the use of IAAFT surrogate time series led us to detect the coexistence of linear and nonlinear dynamics in the day-by-day EF activity monitored at the finer time scale; this result is depicted in Figure 5a where the daily dynamics detected as linear are marked with a black square. When the dynamics are observed on a coarser time scale, we notice that many of the lower IS values measured during periods of restricted mobility in 2020 are associated with linear dynamics, while the corresponding periods in 2019 showing higher IS values also display nonlinear dynamics. This happens for instance comparing the weeks and months covering the population lockdown in spring 2020 with the same periods in 2019 (Figure 5b,c); the effect is documented clearly for the time series of the 4th month shown in Figure 1c and characterized by the IS patterns of Figure 4c: the highly predictable and nonlinear EF dynamics observed in April 2019, resulting from the occurrence of day-night oscillations modulated in amplitude by a slower rhythm with one-week period, are destroyed in April 2020, where the regularity as well as the richness of the dynamics are lost. These findings suggest that both predictability and nonlinearity of the EF intensity time series are lost as a consequence of the restrictions to mobility caused by the COVID-19 pandemic during April and May 2020. A similar effect is observed in August of both years when, due to the summer pause, the campus area is characterized by a lower and less regular presence of university staff and students. Therefore, we conclude that periods characterized as less populated determine not only a decrease of the EF variability, but also an increase of the complexity of the EF patterns and a simultaneous loss of nonlinear dynamics in these patterns.

## 4. Conclusions and Future Perspectives

The results of this study indicate that the EF intensity measured in an urban area with significant movements of people gives rise to time series which can be described as the output of a dynamical system endowed with complex nonlinear and multiscale dynamics. Moreover, the dynamic behavior of the EF intensity can change its properties depending on the temporary accumulation, in proximity of the recording sensor, of EMF sources which can be associated with human utilization of wireless communication systems. Those sources likely refer to mobile telephony systems, including both mobile phones and their base stations installed in the campus area [43], which produce radio-frequency radiation as a means of their communication; these systems integrate phone calls with several features like the Bluetooth and Wi-Fi, which are used by the university population also through other devices like tablets, laptops, and desktop computers. Another example refers to modern vehicles which receive relevant traffic information by using wireless communications from their peers. We find that, in the periods of significantly reduced circulation of university staff and students (e.g., due to summer breaks or to the restrictions to mobility related to the COVID-19 pandemic), not only the amplitude of the oscillations of EF intensity observed at different time scales varied significantly, but also the EF patterns became simultaneously more complex and more linear. These findings document on the one hand the importance of employing non-parametric model-free approaches to fully capture the dynamics of complex systems like this [23], and on the other hand confirm the evidence proven for other types of dynamic systems (e.g., the human heart rate [40]) that complexity and nonlinearity are different, often complementary concepts.

It is worth stressing that the time series of EF variability analyzed here, probing the cumulative intensity of EFs emitted by sources located in the vicinity of the recording sensor, can to some extent be regarded as measurements of human mobility patterns. Hence, our statistical analysis can describe, at the integrated level, the impact of pandemics on human mobility on the territory under study. Different forms of lockdown worldwide, introduced by the Governments to flatten the curve of new infections, affected human mobility both globally and locally on unprecedented scales; the study of implications found in this research can also serve as an opportunity to propose strategies for monitoring and stimulate a more sustainable human mobility. Many recent studies focused on micro-mobility data and analyze the changes in micro-mobility usage before and during the lockdown period exploiting high-resolution micro-mobility trip data [44,45]. From this perspective, our study supports the idea that EF intensity measurements may be complementary to data collections like those from mobile phones, and might be employed to investigate changes in the patterns of mobility both during pandemics or other kinds of events.

A limitation in the exploitation of the data analyzed in this work for the analysis of human mobility may come from the fact these data are recorded in the form of cumulative EF measurements, covering a broad range of frequencies, thus being influenced by EMF sources other than those related to people’s presence and their usage of communication services. This aspect, as well as the possibility of installation of new EMF sources in the campus area, which are likely the main reasons for the presence of mean EF patterns with difficult interpretation observed in some periods of the two analyzed years (Figure 2), have been mitigated in the present study by focusing on measures of variance rather than mean and by performing multiscale analysis of regularity (Figure 3, Figure 4 and Figure 5). Nevertheless, the utilization of frequency-selective measurements may, improve the capability of measures like those proposed in this work to capture a more tight link between human movements and the dynamical patterns of EF intensity.

The interpretation of EF data as time series measuring the temporal variability of the EF intensity can have practical implications also on the assessment of safety related to the spread of EMF sources, and on the plans for expanding the network of sensors in a sustainable and strategic direction. In fact, our approach can be exploited as a supportive tool for the further modeling of network architecture towards more scalable, secure, and cost-effective solutions, which are the requirements of the smart city concept intended as a form of intelligent and sustainable urban development [46]. In this context, future work is envisaged to move towards the study of the joint information shared by multiple EF time series [22], which would open up the possibility of investigating how sensor networks implemented within a territory are functionally interconnected, also providing criteria for the strategic placement/optimization of new sensors both at a local and at a national level.

## Figures and Tables

**Figure 1 entropy-24-00726-f001:**
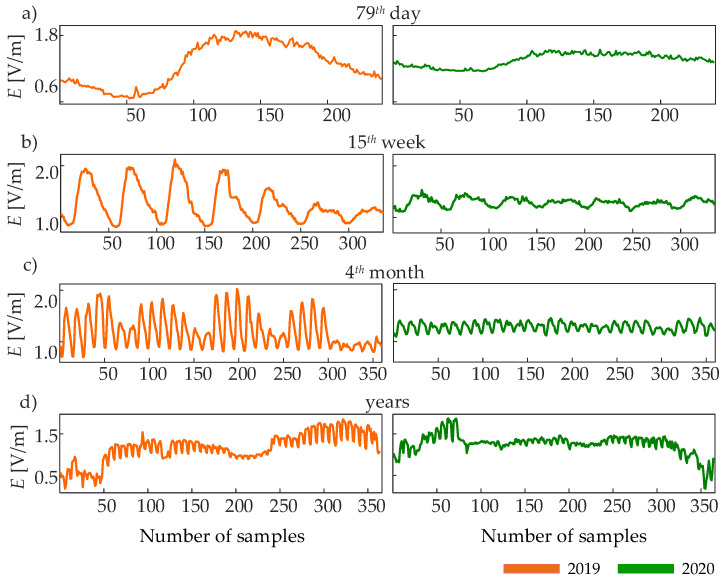
Representative time series of EF intensity monitored during the same day (**a**), week (**b**), and month (**c**) of 2019 (orange) and 2020 (green), as well as during the whole years 2019 and 2020 (**d**). The time series samples are obtained averaging the EF intensity over a time scale that is peculiar of each observation window: $\tau_{day}=6$ min in (**a**); $\tau_{week}=30$ min in (**b**); $\tau_{month}=2$ h in (**c**); $\tau_{year}=1$ day in (**d**).

**Figure 2 entropy-24-00726-f002:**
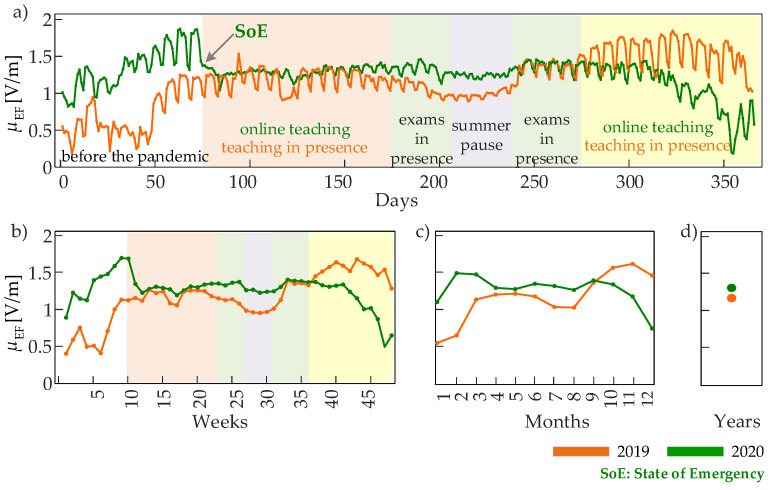
Mean of the EF intensity time series computed over observation windows lasting one day at the time scale $\tau_{day}=6$ min (**a**), one week at the time scale $\tau_{week}=30$ min (**b**), one month at the time scale $\tau_{month}=2$ h (**c**), and one year at the time scale $\tau_{year}=1$ day (**d**). The colored areas identify periods of different activities in the campus area of the University of Novi Sad, occurring before (2019) and during (2020) the COVID-19 pandemic. The colored areas may differ slightly between the two analyzed years (±a few days).

**Figure 3 entropy-24-00726-f003:**
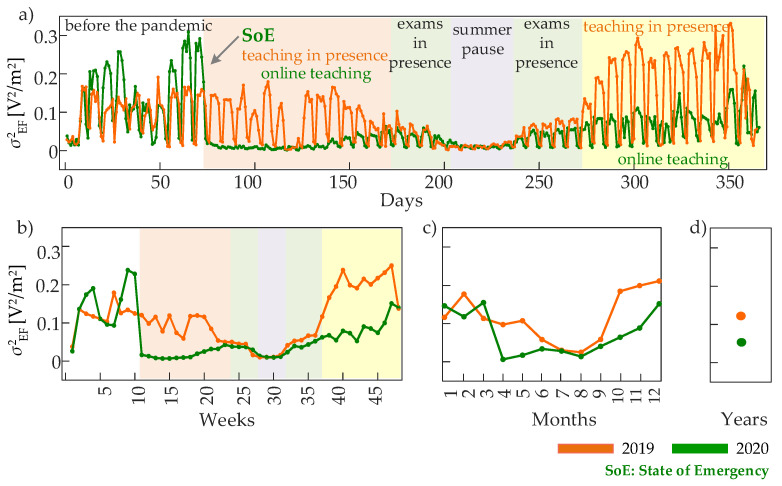
Variance of the EF intensity time series computed over observation windows lasting one day at the time scale $\tau_{day}=6$ min (**a**), one week at the time scale $\tau_{week}=30$ min (**b**), one month at the time scale $\tau_{month}=2$ h (**c**), and one year at the time scale $\tau_{year}=1$ day (**d**). The colored areas identify periods of different activities in the campus area of the University of Novi Sad, occurring before (2019) and during (2020) the COVID-19 pandemic. The colored areas may differ slightly between the two analyzed years (±a few days).

**Figure 4 entropy-24-00726-f004:**
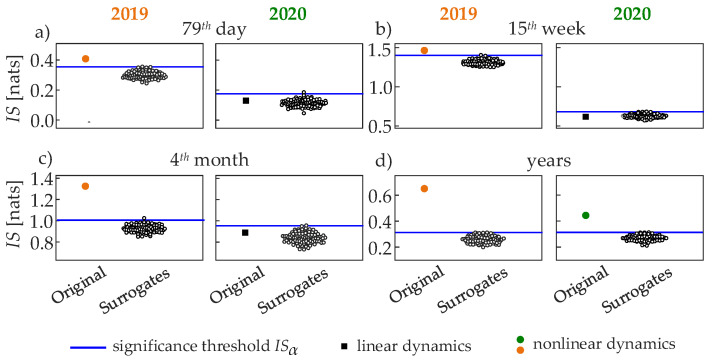
Information storage computed on the representative time series reported in Figure 1 (filled symbols, positioned left) and on 100 IAAFT surrogates (empty circles, right). The thresholds set to detect statistically significant nonlinear dynamics are indicated by blue lines; time series with significant nonlinearity are detected when the original IS exceeds the threshold level (orange or green circles), while the time series is regarded as linear when the original IS is below the threshold (black squares).

**Figure 5 entropy-24-00726-f005:**
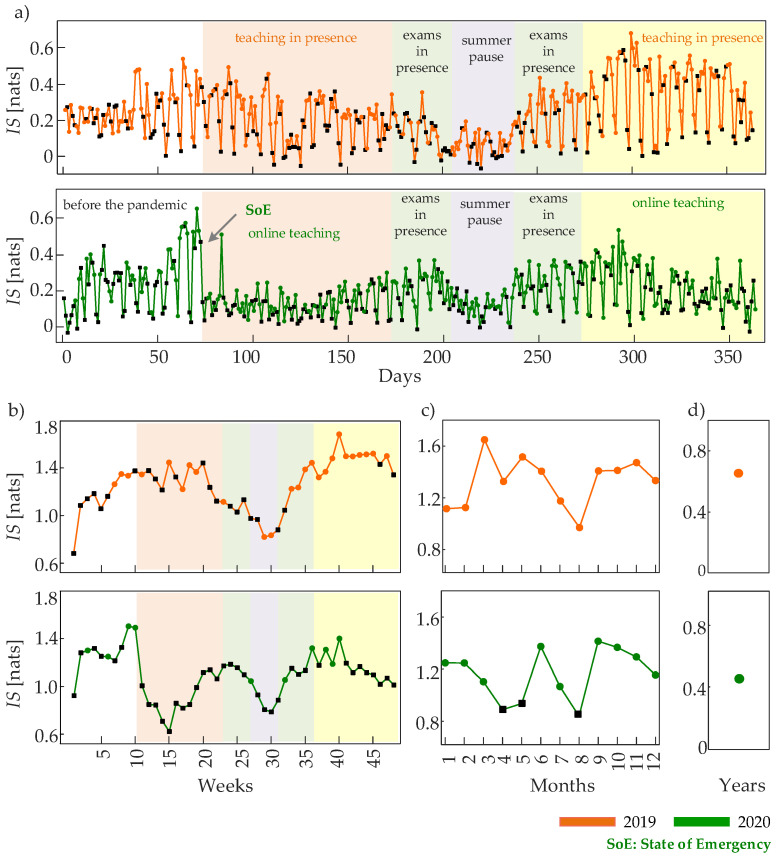
Information storage of the EF intensity time series computed over observation windows lasting one day at the time scale $\tau_{day}=6$ min (**a**), one week at the time scale $\tau_{week}=30$ min (**b**), one month at the time scale $\tau_{month}=2$ h (**c**), and one year at the time scale $\tau_{year}=1$ day (**d**). The colored areas identify periods of different activities in the campus area of the University of Novi Sad, occurring before (2019) and during (2020) the COVID-19 pandemic. The colored areas may differ slightly between the two analyzed years (±a few days). Black-colored squares indicate the presence of linear dynamics, while orange (2019) and green (2020) circles the presence of nonlinear dynamics, detected through the method of surrogate data.

## Data Availability

The data set used here is available and can be freely downloaded from the Serbian Open Data Portal [29]. All analyses were performed using Matlab (The Mathworks, Inc., version R2019b, Natick, MA, USA). The software relevant to the estimation of information storage and generation of IAAFT surrogates is part of the ITS toolbox, which is freely available for download at www.lucafaes.net/its.html, accessed on 3 May 2022.

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
