# Peer review of "Information Dynamics of Electric Field Intensity before and during the COVID-19 Pandemic"

_entropy, 2022, doi:10.3390/e24050726_

Round 1

Reviewer 1 Report

In this work the authors investigate the dynamics of Electric Field  (EF) intensity recorded by the EMF RATEL monitoring system installed in the campus area of the University of Novi Sad, in Serbia, in a two year period ranging from the beginning of 2019, until the end of 2020. This period includes a pre COVID-19 pandemic phase, with regular human activity at the university, but also periods with restricted mobility in the campus, during 2020. The analysis of the data (consisting on average EF intensity measured over consecutive periods of six minutes, and recorded as time series) relies on the Information Storage (IS), which is an information-theoretic measure that relates the regularity of a stochastic process with its predictability, quantifying the amount of information that is shared between the present and the past observations of the stochastic process. To estimate the IS measure, the k-nearest-neighbor-estimator was used, enabling a non-parametric model free approach. The authors present their methodology clearly, referring to relevant and up to date references, and justify the various steps in the analysis with the intent to address the complexity of the data and to fully capture the dynamics of these kinds of systems. Multiple temporal scales were used as the authors performed their IS analysis with four different observation windows lasting one day, one week, one month and one year, with data aggregated in periods of 6min, 30min, 2hours, and 1 day, respectively. They succeed in concluding that both predictability and nonlinearity of the EF time series tend to decrease as a consequence of mobility restrictions and point out that complexity and nonlinearity are different, and often complementary concepts. 

I have a only a few suggestions I would like the Authors to consider, which are the following:

  1. Please correct the typo in line 232 ("...the the...");
  2. I would recommend changing the word "exemplary" in "exemplary time series" (in line 248, for instance, and in Legend of Figure 1, and all other places where it appears) to "representative", as it is a more common scientific expression;
  3. The IS reflects the degree to which information is preserved in a time evolving systems such as any stochastic process. It is nevertheless, a theoretical measure that must be inferred from the data through some estimation technique. As such, I cannot fully agree with a sentence like (Line 328) "The lower values [of IS] document the presence of less predictable, morre erratic EF dynamical patterns...". Care should be taken when drawing these kind of conclusions and I would suggest something like "The lower values are in line with the presence..." or "The lower values point towards the presence...", instead of the sentence mentioned above. There are other situations  in the paper like the one described above, and I would advise a thorough revision to present the proper language in the final version.

Besides these remarks, I have no more suggestions to make, and strongly recommend the publication of the paper.

Author Response

REVIEWER 1

In this work the authors investigate the dynamics of Electric Field (EF) intensity recorded by the EMF RATEL monitoring system installed in the campus area of the University of Novi Sad, in Serbia, in a two year period ranging from the beginning of 2019, until the end of 2020. This period includes a pre COVID-19 pandemic phase, with regular human activity at the university, but also periods with restricted mobility in the campus, during 2020. The analysis of the data (consisting on average EF intensity measured over consecutive periods of six minutes, and recorded as time series) relies on the Information Storage (IS), which is an information-theoretic measure that relates the regularity of a stochastic process with its predictability, quantifying the amount of information that is shared between the present and the past observations of the stochastic process. To estimate the IS measure, the k-nearest-neighbor-estimator was used, enabling a non-parametric model free approach. The authors present their methodology clearly, referring to relevant and up to date references, and justify the various steps in the analysis with the intent to address the complexity of the data and to fully capture the dynamics of these kinds of systems. Multiple temporal scales were used as the authors performed their IS analysis with four different observation windows lasting one day, one week, one month and one year, with data aggregated in periods of 6min, 30min, 2hours, and 1 day, respectively. They succeed in concluding that both predictability and nonlinearity of the EF time series tend to decrease as a consequence of mobility restrictions and point out that complexity and nonlinearity are different, and often complementary concepts. 

We like to thank the reviewer for the careful revision of our paper and for the positive evaluation.

I have a only a few suggestions I would like the Authors to consider, which are the following:

  1. Please correct the typo in line 232 ("...the the...");

The typo is corrected in the revised manuscript.

  1. I would recommend changing the word "exemplary" in "exemplary time series" (in line 248, for instance, and in Legend of Figure 1, and all other places where it appears) to "representative", as it is a more common scientific expression;

We made this change in the revised manuscript.

  1. The IS reflects the degree to which information is preserved in a time evolving systems such as any stochastic process. It is nevertheless, a theoretical measure that must be inferred from the data through some estimation technique. As such, I cannot fully agree with a sentence like (Line 328) "The lower values [of IS] document the presence of less predictable, mor erratic EF dynamical patterns...". Care should be taken when drawing these kind of conclusions and I would suggest something like "The lower values are in line with the presence..." or "The lower values point towards the presence...", instead of the sentence mentioned above. There are other situations in the paper like the one described above, and I would advise a thorough revision to present the proper language in the final version.

Thanks for noting this. We took more care in drawing sharp conclusions rephrasing sentences like the one noticed by the reviewer throughout the whole manuscript.

Besides these remarks, I have no more suggestions to make, and strongly recommend the publication of the paper.

Reviewer 2 Report

Time series measurements of electric field intensity in a university campus throughout the years 2019 and 2020 are studied with the goal of estimating the human activity patterns. Both basic (mean, variance) and advanced (information storage, linearity) analyses are performed. Changes related to COVID-19 lockdowns are detected. The study is a beautiful example of successful reuse of available data. The utiliation of the natural experiment with the lockdown is ingenious.

In general, the paper is written well. However, the methods used are so advanced that there are challenges with the interpretation of the results. As the paper may be interesting for a diverse audience that may not all be familiar with the jargon, it would also be appreciated if some of the terms were more precisely defined. Some parts that raise questions are
- line 41: "microwave ovens" – I do not think they belong among WTT sources.
- line 53: "underlying dynamic nature" – since the term "dynamic" has not been defined, the phrase is distracting.
- line 66: "GPS" – the general term GNSS would be more appropriate as the phones use multiple systems.
- Line 132, reference [28]: please cite the exact dataset and version used. It is important for the reproducibility of the work, and it is appropriate to do so.
- line 157: time scales may be more intuitive to understand if given as time as opposed to inverse time?
- line 163: "complexity and nonlinearity of the dynamic patterns" is quite a phrase, some terms would deserve exact definitions.
- line 194: m=2. Eq. (1) looks at the limit m→∞ – can you comment on why such a small value of m is enough?
- Ew. (3) – can you cite the reference for it?
- same paragraph: "... and is then used to count the number of observations of ..." – the method description is hard to follow for people not already knowing it in detail or unfamiliar with reference [36]. Can you explain it more gradually and precisely, perhaps providing some more formulas?
- line 206: the symbol N' is not actually present in Eq. (5).
- line 222: "realization of a serially uncorrelated process" – does it mean that EF time series would have autocorrelation 0, like if it was a white noise? In this case, it is evident from the time series graphs that EF is highly autocorrelated and the null hypothesis is wrong.
- line 232: "the series is a realization of a Gaussian stochastic process fully described by linear temporal correlations" – could you provide a formula so that we can all agree on what is meant (I like your description, but additional precision would still be appreciated)? Also, the null hypothesis that the activity of the people in the vicinity of the sensor would be a linear process seems unnatural. Can you tell more about the location of the sensor and the human behaviours expected to influence its results? For example, if it is in a quiet part of the park, it will detect the activity of the people in the surrounding buildings (i suppose ...) and an occasional passer-by; if it is within someone's office, it may be predominantly affected by the presence of the worker and his electronic devices in the office.
- line 267 and on: the years 2019 and 2020 are similar in EF mean but not in EF variance. How do you explain that? Were people doing the same things in the same locations both years, resulting in the same mean, but doing it in a time-synchronised manner one year and in a chaotic manner independently of one another next year, decreasing the variance? Or does it have something to do with the nonlinearity between the electric field and the emitted power? EF grows with the square root of the power, so with the square root of the number of people and their devices. Having 10 people in a lecture room for 8 hours and nobody for 16 hours would have the same effect on EF mean as having 1 person around for 24 hours, but the variance would obviously be larger in the first case. Should you have been studying EF squared instead of EF?
- line 281: "the first three months" – the first three months are indeed the period in which 2019 and 2020 differ the most (see Fig. 1), even though there was no epidemic at the time either year. This difference has to be explained, or you cannot claim that you have detected the effect of lockdowns ...
- line 314: would be a good place to add more information about the sensor micro-location and the inferred human behaviours that are detected.
- line 328: "The lower values document the presence of less predictable, more erratic EF dynamical ..." Do you think the interpretation would benefit from a more substantial explanation on what information storage means? Does the difference mean that in 2020, the EF time series was more like noise, while in 2019, it had more of a regular pattern?
- line 336: "complexity and nonlinearity are different concepts" – indeed they are different, and they may be somewhat mutually exclusive in the presented case. See line 186: "H ( X ) = IS ( X ) + CE ( X )". You use IS to measure nonlinearity and CE to measure complexity. If H was constant (I'm not versed enough to know what that would mean), a large complexity CE would mean a small IS and inability to find nonlinearity in it.
- line 344: "resulted always statistically significant in all the analyzed time series" – there is something wrong with the editing of the sentence, and do you basically mean that the time series is not random noise but has some pattern to it?
- line 366: "time series is not significantly different" – personally, I'm not in favour of finding something is not significant and trying to make conclusions about the process from it. You failed to exclude the null hypothesis because you had not enough data or not a good enough method. With more effort, you would statistically significantly detect a difference, no matter how small. By not finding a significant effect, you only get an upper bound on the size of the effect.
- line 410: "GPS, FM Radio" – their emissions are not likely to be affected by the movement of the student population.
Please excuse my ignorance if some of the above comments make no sense.

Thank you for including the Data Availability Statement, but it is not fully compliant. The instructions https://www.mdpi.com/journal/entropy/instructions#suppmaterials require a proper Data citation, and also state: "For work where novel computer code was developed, authors should release the code either by depositing in a recognized, public repository or uploading as supplementary information to the publication. The name and version of all software used should be clearly indicated." The presented analyses were evidently performed by software. Provide it, please.

Author Response

REVIEWER 2

Time series measurements of electric field intensity in a university campus throughout the years 2019 and 2020 are studied with the goal of estimating the human activity patterns. Both basic (mean, variance) and advanced (information storage, linearity) analyses are performed. Changes related to COVID-19 lockdowns are detected. The study is a beautiful example of successful reuse of available data. The utilization of the natural experiment with the lockdown is ingenious.

In general, the paper is written well. However, the methods used are so advanced that there are challenges with the interpretation of the results. As the paper may be interesting for a diverse audience that may not all be familiar with the jargon, it would also be appreciated if some of the terms were more precisely defined.

We like to thank the reviewer for the careful revision of our paper and for the overall positive evaluation.

Some parts that raise questions are

- line 41: "microwave ovens" – I do not think they belong among WTT sources.

We thank the reviewer for this remark. Our intention was to refer to general EMF sources in that place, and therefore we corrected this oversight in the revised manuscript by substituting WTT sources with EMF sources (line 38).

- line 53: "underlying dynamic nature" – since the term "dynamic" has not been defined, the phrase is distracting.

The sentence was rephrased referring in more general terms about the temporal variability of the EF intensity (line 53).

- line 66: "GPS" – the general term GNSS would be more appropriate as the phones use multiple systems.

We thank for this comment; the terminology is now corrected in the revised version of the manuscript (line 65).

- Line 132, reference [28]: please cite the exact dataset and version used. It is important for the reproducibility of the work, and it is appropriate to do so.

We agree with this important observation of the reviewer. In the new manuscript, we reported the link for downloading the CSV file containing the data used in this work (lines 151-152).

- line 157: time scales may be more intuitive to understand if given as time as opposed to inverse time?

The time scales are given as time in the text before that mentioned by the reviewer (see tau values of 6 min, 30 min, 2 hours, 1 day, lines 170-172). However, we agree with the reviewer that the direct mention about scales when referring to the equivalent filter can confuse the reader, and therefore we changed the text referring specifically to the oscillations associated with the time scale and to the frequency of such oscillations (lines 177-180).

- line 163: "complexity and nonlinearity of the dynamic patterns" is quite a phrase, some terms would deserve exact definitions.

According to the comment of the reviewer, and also to clarify the concepts that are defined formally in the following subsections, we provide definitions of complexity and nonlinearity at the beginning of Sect. 2 (lines 108-125).

- line 194: m=2. Eq. (1) looks at the limit m→∞ – can you comment on why such a small value of m is enough?

The approximation of the infinite-length past history of a process with m lagged components corresponds to assume the Markov property for the process, i.e. to assume a memory of m time steps. The assumption m=2 is typical in short-term variability analysis in many contexts, and turns out to be sufficient to describe the regularity of real-world stochastic processes (see, e.g., Refs. 21, 23, 34, 35, 40). Moreover, it takes also into account that high embedding dimensions are difficult to explore in short data realizations, because of the curse of dimensionality that limits the dimension of the spaces for which entropy measures can be reliably computed (see, for instance, Papana, A., Papana-Dagiasis, A., & Siggiridou, E. (2020). Shortcomings of transfer entropy and partial transfer entropy: Extending them to escape the curse of dimensionality. International Journal of Bifurcation and Chaos, 30(16), 2050250.).

- Ew. (3) – can you cite the reference for it?

The reference was added in the revised manuscript (first paragraph, page 6, Ref. [36]).

- same paragraph: "... and is then used to count the number of observations of ..." – the method description is hard to follow for people not already knowing it in detail or unfamiliar with reference [36]. Can you explain it more gradually and precisely, perhaps providing some more formulas?

We explained the method more precisely, describing the steps more gradually and accompanying them with the relevant equations that, summed as in Eq. (4), lead to Eq. (8) (page 6).

- line 206: the symbol N' is not actually present in Eq. (5).

Thanks for noting the inconsistency. Since all the variables were explained previously, we deleted the sentence.

- line 222: "realization of a serially uncorrelated process" – does it mean that EF time series would have autocorrelation 0, like if it was a white noise? In this case, it is evident from the time series graphs that EF is highly autocorrelated and the null hypothesis is wrong

Actually, in statistical hypothesis testing, the null hypothesis cannot be right or wrong, but rather we say that it can be accepted or rejected. In this case, the null hypothesis of an uncorrelated process was stated for the EF time series exactly to test whether this series possesses autocorrelations or not, and in fact, as correctly noted by the reviewer, it was rejected in all the analyzed time series, meaning that the presence of significant autocorrelations was statistically verified using the measure of information storage. This aspect was made clear in the revised paper (lines 360-361, lines 384-386).

- line 232: "the series is a realization of a Gaussian stochastic process fully described by linear temporal correlations" – could you provide a formula so that we can all agree on what is meant (I like your description, but additional precision would still be appreciated)? Also, the null hypothesis that the activity of the people in the vicinity of the sensor would be a linear process seems unnatural.

We enhanced the description of how the surrogate data preserving the linear autocorrelation of the original time series were generated (lines 262-267). We agree with the reviewer that it may be unnatural to assume that the factors determining the EF variability result in a linear process, but actually we do not make this assumption: again, based on statistical hypothesis testing, we formulate a null hypothesis (in this case that the process is linear) and then we test whether this hypothesis is rejected or not; since in many cases the hypothesis is rejected, we indeed conclude that in those cases the process is nonlinear (see, e.g., Figs. 4 and 5). To better recall the philosophy of statistical hypothesis testing, we expanded the initial part of Sect. 2.5 (lines 238-247).

Can you tell more about the location of the sensor and the human behaviours expected to influence its results? For example, if it is in a quiet part of the park, it will detect the activity of the people in the surrounding buildings (i suppose ...) and an occasional passer-by; if it is within someone's office, it may be predominantly affected by the presence of the worker and his electronic devices in the office.

We thank the reviewer for noticing that more details about the location of the sensor would improve the manuscript. The monitoring sensor is installed on the roof of the Faculty of Technical Sciences (FTS), placed in the campus area of University of Novi Sad (the images following the sensor position can be found at the link https://emf.ratel.rs/results/details/eng/9/). Beside the FTS, the campus area encompasses six faculties more, while the FTS presents the largest one by the number of students; currently it has around 16,000 active students. This high number of people ensures a large variability of the overall EF measured by the sensor, which reflects simple day-to-night cycles, weekdays/weekend modulations, then periods of regular teaching and exams compared to holidays and breaks (like the month of August), and even the complete absence of university population induced by the COVID19-lockdown. We added these additional details in the revised manuscript, lines 346-356, as proposed by the reviewer in one of the following points.

- line 267 and on: the years 2019 and 2020 are similar in EF mean but not in EF variance. How do you explain that? Were people doing the same things in the same locations both years, resulting in the same mean, but doing it in a time-synchronised manner one year and in a chaotic manner independently of one another next year, decreasing the variance? Or does it have something to do with the nonlinearity between the electric field and the emitted power?

EF grows with the square root of the power, so with the square root of the number of people and their devices. Having 10 people in a lecture room for 8 hours and nobody for 16 hours would have the same effect on EF mean as having 1 person around for 24 hours, but the variance would obviously be larger in the first case. Should you have been studying EF squared instead of EF?

Our results indicated that the restrictions to human mobility imposed by the pandemic (2020) or the substantial absence of University population during the summer break (during both 2019 and 2020) affected primarily the variability of the electric fields and only to a much lower extent the average field intensity. In the discussion, we stressed that this apparently not-easy interpretation of mean field values observed at all-time scales might be caused by the fact that EF data are recorded in the form of cumulative measurements, covering a broad range of frequencies, thus being influenced by all EMF sources (not just those related to people presence and their usage of communication services). Importantly, we cannot exclude the possibility of installation/detection of new fixed EMF sources that affected the mean field values but had little or no effect on the variability. This observation is added in the revised manuscript, lines 484-485.

Regarding the second part of the comment, we thank the reviewer for this interesting interpretation, which can have an effect on the type of dynamics detected. However, we believe that it would be in general difficult to make deduction about the exact amount of people in the vicinity of the sensor, and thus on the relation between number of persons and EF power, due to the broad-band cumulative nature of the detected field. In this work our goal was less ambitious, i.e. explore how periods with variable concentration of people affect the features of EF intensity considered as the output signal of a dynamic system.

- line 281: "the first three months" – the first three months are indeed the period in which 2019 and 2020 differ the most (see Fig. 1), even though there was no epidemic at the time either year. This difference has to be explained, or you cannot claim that you have detected the effect of lockdowns...

As the reviewer correctly noticed, in the first three months the two observed years differ regarding the mean field values even if one expects to have similar behavior regarding the people presence. As we said before, we conjecture that some new EMF sources (e.g., a mobile-base stations) could have been installed in the beginning of 2020, but this interpretation cannot be underpinned because unfortunately we do not possess exact details. Still, if we observe the COVID year (2020) individually, a sudden drop of mean field is evident in perfect alignment with the start of the lockdown (see arrow “SOE” in Fig. 2a) and a clear reduction in the amplitude of oscillations with one-week period is also correspondingly evident in Fig. 2a. Nevertheless, this basic statistic directed our investigation towards the measures of variance rather than mean, and even more to perform multiscale analysis of regularity, which clearly evidenced the effects of lockdowns as depicted in the following figures.

- line 314: would be a good place to add more information about the sensor micro-location and the inferred human behaviors that are detected.

As suggested, more details about the sensor micro-location are added, page lines 346-356.

- line 328: "The lower values document the presence of less predictable, more erratic EF dynamical ..." Do you think the interpretation would benefit from a more substantial explanation on what information storage means? Does the difference mean that in 2020, the EF time series was more like noise, while in 2019, it had more of a regular pattern?

The interpretation of the reviewer is correct, and was made explicit at this point in the revised paper (lines 108-125). Moreover, in the revised paper, the meaning of the measure of information storage was clarified in the beginning of the methods section (lines 108-125).

- line 336: "complexity and nonlinearity are different concepts" – indeed they are different, and they may be somewhat mutually exclusive in the presented case. See line 186: "H ( X ) = IS ( X ) + CE ( X )". You use IS to measure nonlinearity and CE to measure complexity. If H was constant (I'm not versed enough to know what that would mean), a large complexity CE would mean a small IS and inability to find nonlinearity in it.

Actually the two concepts are not necessarily mutually exclusive; complexity is measured by CE(X) in a direct way, and by IS(X) in a complementary way (higher complexity results in higher CE and lower IS); on the other hand, nonlinearity refers to the type of dynamics, and is tested using surrogate data and computing either CE or IS (in our work we compute IS). To clarify these concepts and favor the interpretation of the results in view of them, we added an explanatory paragraph at the beginning of the method section in the revised paper (lines 108-125).

- line 344: "resulted always statistically significant in all the analyzed time series" – there is something wrong with the editing of the sentence, and do you basically mean that the time series is not random noise but has some pattern to it?

Yes, we mean that the time series has some regular patterns and this is identified finding statistically significant values of the IS. The sentence was rewritten (lines 384-386), see also our reply to the previous comment "- line 222: realization of a serially uncorrelated process".

- line 366: "time series is not significantly different" – personally, I'm not in favour of finding something is not significant and trying to make conclusions about the process from it. You failed to exclude the null hypothesis because you had not enough data or not a good enough method. With more effort, you would statistically significantly detect a difference, no matter how small. By not finding a significant effect, you only get an upper bound on the size of the effect.

Since the considerations in this part of the discussion were not actually based on the statistical tests, we rephrased the sentence avoiding mention to “significant differences” (lines 408-409).

- line 410: "GPS, FM Radio" – their emissions are not likely to be affected by the movement of the student population.

Our intention was to refer to radio-frequency emission as a means of mobile phones – their base stations communication, when students in general exploit mobile telephony system due to various services than can be provided (for example, not just phone calls and Wi-Fi, but also listening radio by using mobile-phone). In order to avoid misleading interpretations, we deleted this in the revised manuscript, line 452.

Please excuse my ignorance if some of the above comments make no sense.

We sincerely thank the reviewer for the many constructive comments, which also urged us to better clarify those parts of the paper which are not easy to grasp for the non-expert reader.

Thank you for including the Data Availability Statement, but it is not fully compliant. The instructions https://www.mdpi.com/journal/entropy/instructions#suppmaterials require a proper Data citation, and also state: "For work where novel computer code was developed, authors should release the code either by depositing in a recognized, public repository or uploading as supplementary information to the publication. The name and version of all software used should be clearly indicated." The presented analyses were evidently performed by software. Provide it, please.

All analyses were performed using MATLAB (The Mathworks, Inc., version R2019b). The software relevant to the estimation of information storage and generation of IAAFT surrogates is part of the ITS toolbox, which is freely available for download at //www.lucafaes.net/its.html. This information is now provided in the revised manuscript, lines 515-518.

Round 2

Reviewer 2 Report

Perfect!

I have no further meaningful suggestions, except maybe for the lines 246-247, where you say "... detect the presence of the investigated property if the null hypothesis is rejected, or the absence of the property if the null hypothesis is not rejected ..." I philosophically disagree and would suggest you state it a bit differently (failing to reject the null hypothesis does not mean that the null hypothesis is correct at all). However, this is just a suggestion, I wouldn't call it a need for "revision".